# Rapid Identification of Paeoniae Radix and Moutan Radicis Cortex Using a SCAR Marker-Based Conventional PCR Assay

**DOI:** 10.3390/plants11212870

**Published:** 2022-10-27

**Authors:** Wook Jin Kim, Sumin Noh, Goya Choi, Byeong Cheol Moon

**Affiliations:** Herbal Medicine Resources Research Center, Korea Institute of Oriental Medicine, Naju 58245, Korea

**Keywords:** Paeoniae Radix, Moutan Radicis Cortex, *Paeonia* species, sequence characterized amplified region (SCAR), PCR assay, herbal medicine

## Abstract

Paeoniae Radix is a herbal medicine prepared from the dried roots of *Paeonia lactiflora*, *P. anomala* subsp. *veitchii*, and *P. japonica*. Although the herbal medicines prepared from these species are morphologically similar, they have different pharmacological effects depending on how they are processed. In addition, *P. japonica* is more expensive than other *Paeonia* spp. in the Korean herbal market. Although there is a clear difference between the Korean and Chinese pharmacopeias of Paeoniae Radix, the processed roots of *P. lactiflora* and *P. anomala* subsp. *veitchii* are commonly used indiscriminately in the herbal market. Moreover, *Paeonia suffruticosa*, an allied genus of *P. lactiflora*, is prescribed as Moutan Radicis Cortex. Therefore, accurate taxonomic identification of plant species is vital for quality assurance. A genetic assay is a reliable tool for accurately discriminating species in processed herbal medicines. To develop a genetic assay for the identification of four *Paeonia* species (*P. lactiflora*, *P. anomala* subsp. *veitchii*, *P. japonica*, and *P. suffruticosa*), we analyzed the sequences of two DNA barcoding regions, internal transcribed spacer and *rbc*L. A conventional PCR assay was established in this study for simple and rapid species identification using sequence characterized amplified region (SCAR) markers based on arbitrary nucleotide-containing primers. This assay was verified to be species specific and highly sensitive and could be applied to *Paeonia* species identification at an affordable rate.

## 1. Introduction

Paeoniae Radix (PR) is a herbal medicine commonly used in Asian countries, especially China and Korea, for curing extravasated blood, relieving pain, alleviating fever, and inducing hepatoprotection [1,2]. *Paeonia*
*lactiflora* Pall. and its closely related species are authentic sources of PR, according to the Korean Pharmacopoeia [1]. However, three *Paeonia* species (*P. lactiflora*, *P. japonica* (Makino) Miyabe & Takeda, and *Paeonia anomala* subsp. *veitchii* (Lynch) D.Y.Hong & K.Y.Pan) are frequently distributed as PR in the Korean herbal market (Appendix A) [3,4]. In China, an important exporting country of PR to Korea, two herbal medicines are prepared from the roots of *P. lactiflora*, depending on the method of processing: Paeoniae Radix Rubra (PRR) and Paeoniae Radix Alba (PRA) [1,3]. Thus, PRR is defined as the dried root of two species (*P. lactiflora* and *P. anomala* subsp. *veitchii*) according to the Pharmacopoeia of the People’s Republic of China and is usually prescribed for the same pharmacological purpose as Korean PR [1,5]. On the other hand, PRA is defined as only the peeled root of *P.*
*lactiflora* and is usually prescribed for its anti-inflammatory and immune regulatory effects in Chinese Medicine [1,5]. Additionally, the root bark of *Paeonia suffruticosa* is used as the source of Moutan Radicis Cortex (MRC), a herbal medicine used to activate blood, but is not used as a source of PR [1,6].

Most herbal medicines containing one or more species are considered authentic [1,7]. Moreover, some herbal medicines containing species belonging to different genera are used as substitutes for each other, because they have equivalent pharmacological effects [1,7]. Medicinal plants present many possibilities for the development of new drugs and dietary supplements as they are inherently abundant in natural health-promoting compounds [8,9]. The pharmacological and chemical characteristics of *Paeonia* species are also studied at the species level [7,10,11].

Herbal medicines are frequently contaminated by adulterants due to such reasons as economic motivation, morphological similarity, and synonymous names [12,13]. In Korea, PR herbal medicine and *P. lactiflora* are called *Jak-yak*, which is a synonym. Because of its scarcity, *P. japonica* is more expensive than the other *Paeonia* spp. (*P. lactiflora* and *P. anomala* subsp. *veitchii*) in the Korean herbal market [4]. The peeled root of *P. lactiflora* is sold as *P. japonica* in the Korean herbal market because of the morphological similarity between the two species and the economic motivation of the seller. Therefore, accurate species identification is vital for the quality control of raw materials. To prevent the contamination of herbal medicines, the development of a diagnostic assay is required.

Species identification uses multidisciplinary approaches such as morphological evaluation, chemical analysis, and genetic assay [1,3,4]. Among these approaches, a genetic assay is cheap and guarantees accurate results without requiring highly skilled personnel and expensive analytic instruments [13]. Recently, genomic regions such as internal transcribed spacer (ITS), *mat*K, *rbc*L, and *psb*A-*trn*H have been used as DNA barcodes for the species-level identification of herbal medicines [14,15]. However, DNA barcoding analysis is not effective in all taxa because of no PCR amplification or low amplification efficiency, which makes it difficult to analyze the sequence [15]. Moreover, most herbal medicines are distributed in a sliced and/or dried state, which makes it difficult to trace some DNA barcoding regions, such as the 1.3 kb *mat*K and 1.6 kb *rbc*L [16].

A genetic assay involving conventional PCR, DNA barcoding, and sequence characterized amplified region (SCAR) marker-based genotyping is highly advantageous, since it can omit cumbersome processes such as gel extraction and sequence analysis [7]. Therefore, compared with DNA barcoding analysis, the conventional PCR assay is cheap and offers rapid results in 1 day [13].

In this study, we sequenced the ITS and *rbc*L regions of *P. lactiflora*, *P. japonica*, *P. anomala* subsp. *veitchii*, and *P. suffruticosa*, and compared their sequences to determine interspecific variation. The discriminability and sensitivity of SCAR primers were validated using conventional PCR. Moreover, the SCAR markers developed in this study could accurately identify commercial PR and MRC herbal medicines.

## 2. Results

### 2.1. DNA Barcoding Analysis

Two DNA barcoding regions, ITS and *rbc*L, were analyzed in 35 plant samples (Appendix A). Among the 35 samples used in this study, 17 have been sequenced previously [4]. Raw sequences of the remaining 18 samples newly reported in this study were obtained from the T7 and SP6 promoters located in the T-vector and processed by BioEdit to eliminate unnecessary sequences. Then, sequences of the analyzed region were registered in the GenBank database of the National Center for Biotechnology Information (NCBI) (see Appendix A for accession numbers). The amplicon length of the ITS region was 743 nt in all four *Paeonia* species (Appendix A), while that of *rbc*L was 1511 nt in three species (*P. lactiflora*, *P. anomala* subsp. *veitchii*, and *P. suffruticosa*) and 1530 nt in *P. japonica* (Appendix A).

### 2.2. SCAR Primer Design and PCR Assay Verification

Species-specific SCAR primers for the conventional PCR assay were designed in the sequence showing interspecific variability and intraspecific conservation (Table 1). SCAR primers were designed in the ITS region for three species (*P. japonica*, *P. anomala* subsp. *veitchii*, and *P. suffruticosa*) and in the *rbc*L region for *P. lactiflora*. An arbitrary nucleotide was used at the 3′ end of the forward and reverse primers for enhancing sequence specificity, namely, PL-F (T → C), PJ-F (G → T), PJ-R (C → G), PA-R (C → G), and PS-F (G → C), where the abbreviations PL, PJ, PA, and PS represent the four species (*P. lactiflora*, *P. japonica*, *P. anomala* subsp. *veitchii*, and *P. suffruticosa*, respectively), and F and R indicate forward and reverse primers, respectively. The melting temperature of the SCAR primer was designed to be approximately 55 °C.

To develop the conventional PCR assay, we tested the species specificity of candidate SCAR primers in 16 different combinations for *P. lactiflora*, 59 combinations for *P. japonica*, 8 for *P. anomala* subsp. *veitchii*, and 8 for *P. suffruticosa* (data not shown). Then, gradient PCRs at temperatures ranging from 55 °C to 65 °C were performed using the selected primer combinations (data not shown). Consequently, the annealing temperature of species-specific SCAR primers was determined to be 63 °C for all four species. Figure 1 shows the results of PCR amplification performed under optimal conditions using the finally selected SCAR primers. A 430 bp fragment was amplified from all ten accessions of *P. lactiflora* with the PL-F and PL-R primer combination, but no amplification was detected in nontarget species. Similarly, other SCAR primer combinations (PJ-F/-R for *P. japonica*, PA-F/-R for *P. anomala* subsp. *veitchii*, and PS-F/-R for *P. suffruticosa*) also showed species specificity (Figure 1).

The selected SCAR primer combinations were also tested against 21 other plant species used to prepare herbal medicines, including *Panax ginseng*, *Glycyrrhiza uralensis*, *Cnidium officinale*, *Angelica gigas*, *Schisandra chinensis*, *Zanthoxylum schinifolium*, *Aralia continentalis*, *Cynanchum wilfordii*, *Angelica dahurica*, *Akebia quinata*, *Viscum coloratum*, *Adenophora stricta*, *Sigesbeckia orientalis* subsp. *pubescens*, *Rheum rhabarbarum*, *Machilus thunbergii*, *Scutellaria baicalensis*, *Clematis terniflora* var. *mandshurica*, *Prunus armeniaca* var. *ansu*, *Trichosanthes kirilowii*, *Ulmus macrocarpa*, and *Prunella vulgaris* subsp. *asiatica* (Table 2). The results confirmed the species specificity of the SCAR primers (Appendix A and Table 2).

### 2.3. Sensitivity and Discriminability of SCAR Primers

The sensitivity of the selected SCAR primer combinations was evaluated by measuring the limit of detection (LOD) using serial dilutions of the DNA template. The LOD value of the SCAR primer pairs for *P. lactiflora*, *P. anomala* subsp. *veitchii*, and *P. suffruticosa* (PL-F/-R, PA-F/-R, and PS-F/-R, respectively) was 10 pg, and that of the primer pair for *P. japonica* (PJ-F/-R) was 100 pg (Figure 2). In addition, the selected SCAR primer pairs could accurately identify the constituents of a mixed DNA sample (Figure 3).

### 2.4. Monitoring of Commercial PR and MRC Samples

We determined the composition of a total of 21 commercial products (16 PR and 5 MRC) using morphological and genetic assays (Table 3 and Figure 4). Among the 16 products commercially labeled as PR, 10 were identified as *P. lactiflora*, four as *P. japonica*, and two as *P. anomala* subsp. *veitchii* based on morphological analysis (Appendix A), while 12 were identified as *P. lactiflora*, 3 as *P. japonica*, and 1 as *P. anomala* subsp. *veitchii* based on the SCAR marker assay. On the other hand, all five commercial MRC samples were identified as *P. suffructicosa* in both the morphological and SCAR marker assays (Figure 4). Additionally, the results of DNA barcoding analysis using the ITS region were consistent with those of the SCAR marker assay (Table 3 and Appendix A).

## 3. Discussion

Herbal medicines can be easily mislabeled for economic gain economic because authentic species and adulterants are morphologically similar and have similar names [12,13]. In traditional Korean medicine, *P. lactiflora* is considered an authentic herbal medicine (PR) species, together with *P. japonica* and *P. anomala* subsp. *veitchii* [1,3,4]. However, because of economic motivation, *P. lactiflora* is sometimes traded as *P. japonica*, which is more expensive than the other *Paeonia* species in the Korean herbal market [4]. PRA, the peeled root of *P. lactiflora*, is called *Baek-jak* in Korean, which is synonymous with *Baek-jak-yak*, a herbal medicine derived from *P. japonica* [4]. Therefore, PRA could possibly be mislabeled as *P. japonica* [4]. Moreover, the herbal medicines PR and MRC are distributed in processed forms such as slices and dried roots, which makes it difficult to identify the plant species based on morphological evaluation [4]. Therefore, highly trained professionals, such as herbologists and taxonomists, are needed for the accurate morphological identification of species [3]. In recent studies, artificial intelligence using the deep-learning model has been employed for the morphological discrimination among species; however, this platform is not yet complete and therefore not reliable [17].

Recently, pharmacological analyses have been conducted to identify the effective components of *Paeonia* species [7,10,11]. Shi et al. reported that *P. lactiflora* and *P. anomala* subsp. *veitchii* contain more monoterpenoids than other *Paeonia* species [5]. Monoterpenoids, the main bioactive compounds unique to *Paeonia* species, possess anti-inflammatory, antioxidant, anticoagulative, antiallergic, sedative, and analgesic properties [18,19]. Ko et al. reported that *P. japonica* contains many bioactive compounds such as albiflorin, paeoniflorigenone, oxybenzoyl-paeoniflorin, paeonol, and paeoniflorin [10]. Therefore, we may discover new natural compounds with excellent pharmacological properties among these bioactive constituents.

The importance of diagnostic assay development was realized in Korea in 2015 during a confusing market situation called the *Baek-suo* situation. *Cynanchum wilfordii*, an authentic component of Cynanchi Wilfordii Radix, was used to synthesize dietary supplements and herbal medicines; however, *C. wilfordii* was replaced by *Cynanchum auriculatum*, a toxic adulterant, in these products [20,21]. However, the confusion soon came to an end when previously developed diagnostic PCR assays of *C. wilfordii* and *C. auriculatum* were used to test the adulterated products. Therefore, developing diagnostic assays preemptively can help avoid market confusion.

Genetic studies have been conducted on the PR herbal medicine and *Paeonia* species, and DNA barcoding and peptide nucleic acid (PNA) probe assays have been reported [4]. Conventional PCR assay using species-specific primers is faster, cheaper, and more accurate than DNA barcoding analysis [7,22]. Kim et al. reported the development of a peptide nucleic acid (PNA) probe-based real-time PCR assay capable of rapidly and accurately discriminating among four *Paeonia* species, since the PNA probe could detect single nucleotide polymorphisms (SNPs) [4]. However, this assay requires expensive real-time PCR instruments with multichannel specifications, as well as a specially designed probe [4]. Moreover, this assay could discriminate only among three species, not among all four *Paeonia* species [4].

Therefore, we designed SCAR primers based on the sequences of the two most frequently used DNA barcodes (ITS and *rbc*L) and developed a simple and low-cost genetic assay that could be performed on a conventional PCR machine. Firstly, we analyzed the DNA barcoding regions to confirm that the ITS and *rbc*L regions could be discriminated. The species-specific SCAR primers were established by performing a singleplex assay on a conventional PCR machine. An arbitrary nucleotide was included at the 3′ terminal end of the primer and was confirmed to enhance the specificity of PCR amplification [22,23]. Moreover, we considered a small amplicon size tractable for the SCAR assay because this assay is applied to physically degraded and modified herbal medicines with poor amplification efficiency. The species-specific SCAR primers were developed in the ITS region for all species, except *P. lactiflora*; the ITS region of *P. lactiflora* could not be used for designing SCAR primers because it did not contain species-specific nucleotide variation. Instead, the *rbc*L region of *P. lactiflora* was used for SCAR primer design, and a singleplex assay was successfully established. The specificity of these SCAR primers was validated against 21 commercial herbal medicines (16 PR and 5 MRC), which are the most frequently prescribed in traditional Korean medicine. Moreover, these SCAR primers showed high sensitivity, with LOD scores ranging from 10 to 100 pg [22,23]. Finally, we confirmed the mislabeled products among the 21 commercial samples.

The genetic assay developed in this study is anticipated to be a useful technique for the standardization and quality control of PR and MRC herbal medicines. Furthermore, this assay could be used for the rapid identification of closely related species such as *Paeonia* species.

## 4. Materials and Methods

### 4.1. Plant Material and DNA Extraction

Thirty-five samples belonging to four species, *P. lactiflora*, *P. japonica*, *P. anomala* subsp. *veitchii*, and *P. suffruticosa*, were used in this study. These samples were collected from South Korea and China, and specimens identified by plant taxonomists and ecologists were deposited in the KIOM herbarium (index herbarium [IH] code: KIOM). Genomic DNA was extracted using the DNeasy^®^ Plant Mini Kit (Qiagen, Valencia, CA, USA), according to the manufacturer’s instructions. DNA quality and quantity were measured using an ND-1000 UV/Vis spectrophotometer (NanoDrop, Wilmington, DE, USA).

### 4.2. DNA Barcoding Analysis

The ITS region was amplified using ITS1 (5′-TCC GTA GGT GAA CCT GCG G-3′) and ITS4 (5′-TCC TCC GCT TAT TGA TAT GC-3′) primers, and the *rbc*L region was amplified using the rbcL-F (5′-ATG TCA CCA CAA ACA GAA ACT AAA GC-3′) and rbcL-R (5′-TCC TTT TAG TAA AAG ATT GGG CGG AG-3′) primers [24,25]. Each PCR reaction contained 0.5 µM forward and reverse primers, Solg^TM^ 2× Taq PCR Smart-Mix I (Solgent, Daejeon, Korea), and approximately 10 ng of DNA template. PCR conditions are as follows: initial denaturation at 95 °C for 2 min; 35 cycles of denaturation at 95 °C for 1 min, annealing at 53 °C for 1 min, and extension at 72 °C for 2 min; and a final extension at 72 °C for 5 min. Subsequent experiments were performed as described previously. Sequence information of 35 samples was analyzed by ClustalW using the BioEdit software version 7.2.5 (Raleigh, NC, USA) created by Thomas A. Hall.

### 4.3. SCAR Primer Design and PCR Assay Verification

Primers were designed based on the nucleotide sequence variation among species. Species-specific primers were designed in the ITS region for *P. japonica*, *P. anomala* subsp. *veitchii*, and *P. suffruticosa*, and in the *rbc*L region for *P. lactiflora*. PCR reactions were prepared as described in Section 4.2. To determine the optimal annealing temperature, gradient PCR was performed at 55–65 °C. Then, using 63 °C as the optimal annealing temperature, PCR was performed as follows: initial denaturation (95 °C for 2 min); 35 cycles of denaturation, annealing, and extension (95 °C for 30 s, 63 °C for 30 s, and 72 °C for 30 s, respectively); and final extension (72 °C for 5 min). Experimental verification for species specificity was carried out in three replicates. Twenty-one herbal medicine-related plant species (Table 2) were tested to verify the specificity of the SCAR primer pairs.

### 4.4. Sensitivity and Discriminability of SCAR Primers

The sensitivity of SCAR primers was analyzed using 10-fold serial dilutions (10 ng–10 fg) of the genomic DNA of the target species. Discriminability was evaluated using mixed DNA samples, which were prepared by mixing the ground leaf tissue of two to four species (1 g each), followed by DNA extraction as described in Section 4.1. PCR reaction composition and thermocycling conditions were the same as described in Section 4.3. Experimental verification for sensitivity was carried out in three replicates.

### 4.5. Monitoring of Commercial PR and MRC Samples

A total of 21 commercial products (16 PR and 5 MRC) were analyzed in this study. Identification based on morphological evaluation was performed by herbologists. Genetic evaluation was conducted by performing the SCAR assay and DNA barcoding analysis. Briefly, 10 g of each sample was subjected to DNA extraction as described above (Section 4.1), and PCR was performed as described in Section 4.3. Finally, DNA barcoding analysis was carried out using the ITS region.

## Figures and Tables

**Figure 1 plants-11-02870-f001:**
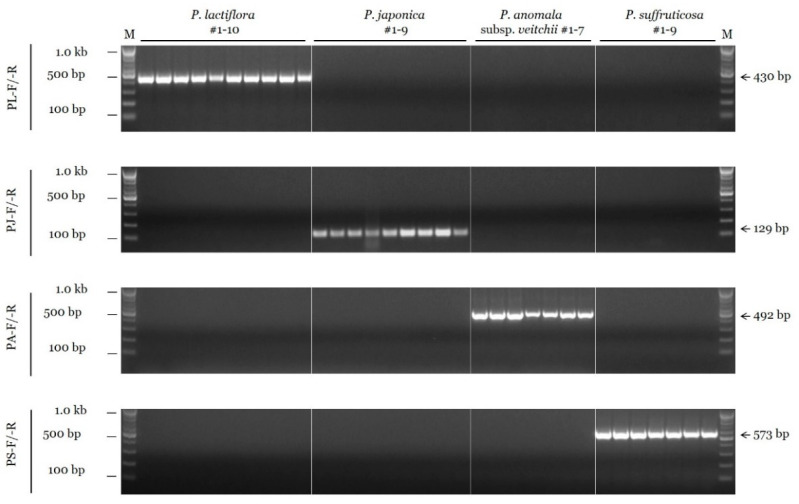
Specificity of the SCAR primers using the conventional PCR assay. SCAR primer combinations (PL-F/-R, PJ-F/-R, PA-F/-R, and PS-F/-R) are shown on the left. ‘M’ indicates the 100 bp DNA ladder, and the sizes of different bands (100 bp, 500 bp, and 1.0 kb) are indicated on the left. Arrows and numbers on the right indicate the amplicon size.

**Figure 2 plants-11-02870-f002:**
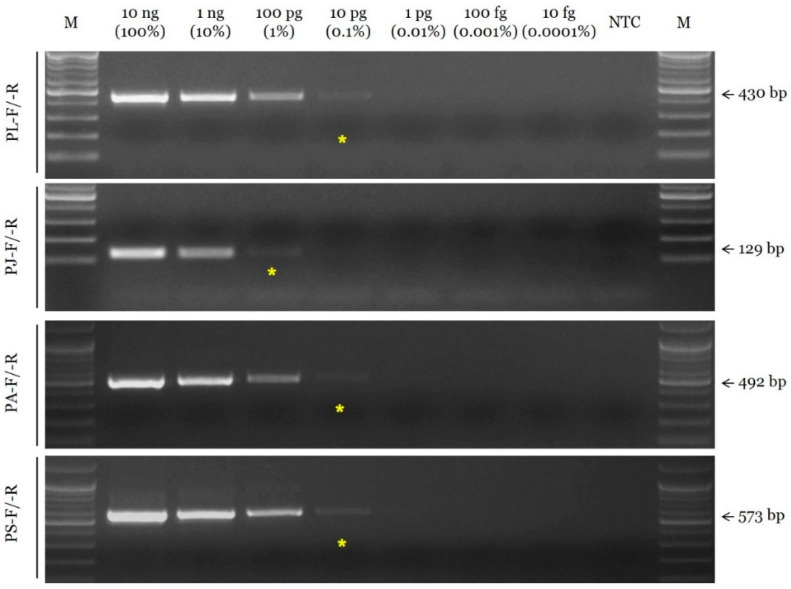
Assessment of the sensitivity of SCAR primers using the conventional PCR assay. Primer combinations (PL-F/-R, PJ-F/-R, PA-F/-R, and PS-F/-R) are indicated on the left. Arrows and numbers situated on the right indicate the amplicon size. M is the 100 bp DNA ladder. Yellow asterisks indicated the limit of detection (LOD).

**Figure 3 plants-11-02870-f003:**
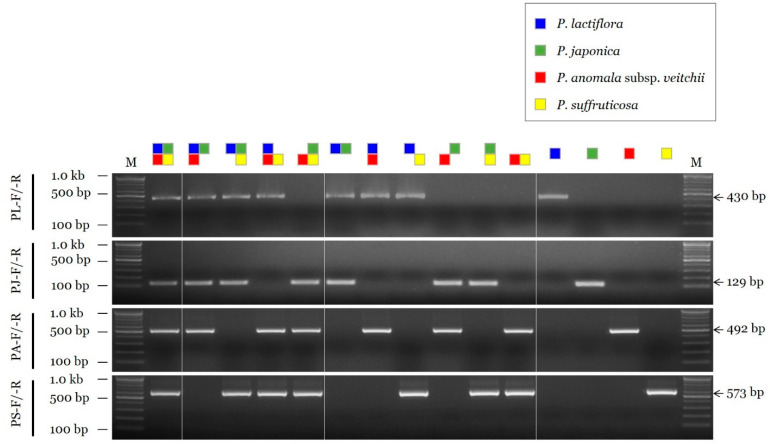
Assessment of the discriminability of SCAR primers using mixed DNA samples. The composition of each DNA sample is shown on the top using different colored squares (blue, *P. lactiflora*; green, *P. japonica*; red, *P. anomala* subsp. *veitchii*; yellow, *P. suffruticosa*). SCAR primer combinations (PL-F/-R, PJ-F/-R, PA-F/-R, and PS-F/-R) and 100 bp, 500 bp, and 1.0 kb bands of the 100 bp DNA marker (M) are indicated on the left. Arrows and numbers on the right indicate the amplicon size.

**Figure 4 plants-11-02870-f004:**
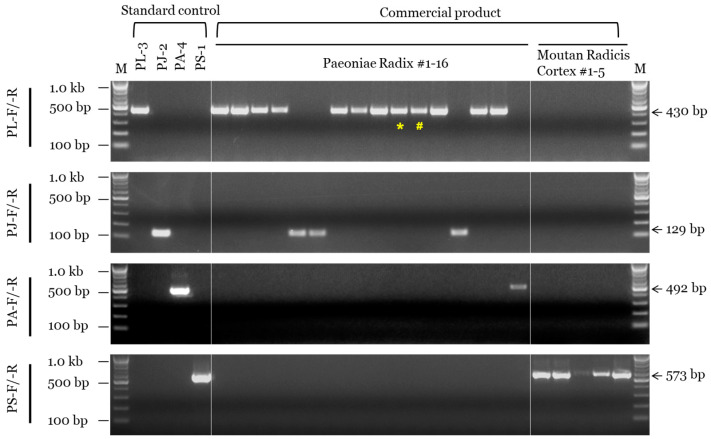
SCAR assay of PR and MRC samples sold in the Korean herbal market. SCAR primer combinations (PL-F/-R, PJ-F/-R, PA-F/-R, and PS-F/-R) and 100 bp, 500 bp, and 1.0 kb bands of the 100 bp DNA marker (M) are indicated on the left. List information is provided in Appendix A and Table 3. Arrows and numbers on the right indicate amplicon size. Special characters (* and #) indicate inconsistencies between the results of morphological and genetic assays.

**Table 1 plants-11-02870-t001:** Details of primers used to perform the conventional PCR assay.

Primer Name	Target Species	Primer Sequence (5′→3′) ^1^	Amplicon Size (bp)	DNA Barcoding Region Used for SCAR Primers
PL-F	*P. lactiflora*	TTT CTT ATT TTG TGC AGA AGC **C**C	430	*rbc*L
PL-R		ACG ACT TCG GTC TTT TTC AAT AT		
PJ-F	*P. japonica*	GAA CTT GTA AAA ATG CTC GG**T** G	129	ITS
PJ-R		CCG AGA GCA CAC CAG AA**G** G		
PA-F	*P. anomala* subsp. *veitchii*	AAG GCG TGA GCC TCT CCT C	492	ITS
PA-R		GTT CGT CGC TCG GGG CT**G** A		
PS-F	*P. suffruticosa*	CTC CTT CAT CCC ACG TCC **C**A	573	ITS
PS-R		TCG TGA CGA TGC TCT GGG GTT		

^1^ Bold underlined character indicate the arbitrary sequence.

**Table 2 plants-11-02870-t002:** Analysis of the specificity of SCAR primers using 21 herbal medicine-related plant species.

Herbal Medicine	Plant Species	DNA Amplification ^1^			
		PL-F/-R	PJ-F/-R	PA-F/-R	PS-F/-R
Paeoniae Radix	*Paeonia lactiflora*	+	–	–	–
Paeoniae Radix	*Paeonia japonic* *a*	–	+	–	–
Paeoniae Radix	*Paeonia anomala* subsp. *veitchii*	–	–	+	–
Moutan Radicis Cortex	*Paeonia suffruticosa*	–	–	–	+
Ginseng Radix	*Panax ginseng*	–	–	–	–
Glycyrrhizae Radix et Rhizoma	*Glycyrrhiza uralensis*	–	–	–	–
Cnidii Rhizoma	*Cnidium officinale*	–	–	–	–
Angelicae Gigantis Radix	*Angelica gigas*	–	–	–	–
Schisandra Fruit	*Schisandra chinensis*	–	–	–	–
Zanthoxylum Peel	*Zanthoxylum schinifolium*	–	–	–	–
Aralia Continentalis Root	*Aralia continentalis*	–	–	–	–
Cynanchi Wilfordii Radix	*Cynanchum wilfordii*	–	–	–	–
Angelica Dahurica Root	*Angelica dahurica*	–	–	–	–
Akebia Stem	*Akebia quinata*	–	–	–	–
Visci Ramulus et Folium	*Viscum coloratum*	–	–	–	–
Adenophorae Radix	*Adenophora stricta*	–	–	–	–
Siegesbeckiae Herba	*Sigesbeckia orientalis* subsp. *pubescens*	–	–	–	–
Rhei Undulatai Rhizoma	*Rheum rhabarbarum*	–	–	–	–
Magnoliae Cortex	*Machilus thunbergii*	–	–	–	–
Scutellariae Radix	*Scutellaria baicalensis*	–	–	–	–
Clematidis Radix	*Clematis terniflora* var. *mandshurica*	–	–	–	–
Armeniacae Semen	*Prunus armeniaca* var. *ansu*	–	–	–	–
Trichosanthis Semen	*Trichosanthes kirilowii*	–	–	–	–
Ulmi Cortex	*Ulmus macrocarpa*	–	–	–	–
Prunellae Spica	*Prunella vulgaris* subsp. *asiatica*	–	–	–	–

^1^ ‘+’ Indicates amplicon detected; ‘–’ indicates no amplification.

**Table 3 plants-11-02870-t003:** Morphological and genetic evaluation of herbal medicines sold as PR and MRC in the Korean herbal market.

Herbal Medicine	Sample No.	Voucher No.	Sample Identity Based on Different Assays ^1^		
			Morphological Assay	SCAR Assay	DNA Barcoding Assay of the ITS Region
PR	1	2-19-0431	*P. lactiflora*	*P. lactiflora*	*P. lactiflora*
	2	2-17-0572	*P. lactiflora*	*P. lactiflora*	*P. lactiflora*
	3	2-17-0358	*P. lactiflora*	*P. lactiflora*	*P. lactiflora*
	4	2-17-0035	*P. lactiflora*	*P. lactiflora*	*P. lactiflora*
	5	2-16-0174	*P. japonica*	*P. japonica*	*P. japonica*
	6	2-16-0124	*P. japonica*	*P. japonica*	*P. japonica*
	7	2-15-0201	*P. lactiflora*	*P. lactiflora*	*P. lactiflora*
	8	2-14-0200	*P. lactiflora*	*P. lactiflora*	*P. lactiflora*
	9	2-14-0017	*P. lactiflora*	*P. lactiflora*	*P. lactiflora*
	10	2-12-0114	** *P. japonica* **	** *P. lactiflora* **	** *P. lactiflora* **
	11	2-12-0105	***P. anomala*** **subsp. *veitchii***	** *P. lactiflora* **	** *P. lactiflora* **
	12	2-12-0032	*P. lactiflora*	*P. lactiflora*	*P. lactiflora*
	13	2-12-0010	*P. japonica*	*P. japonica*	*P. japonica*
	14	2-11-0130	*P. lactiflora*	*P. lactiflora*	*P. lactiflora*
	15	2-11-0073	*P. lactiflora*	*P. lactiflora*	*P. lactiflora*
	16	2-07-0034	*P. anomala* subsp. *veitchii*	*P. anomala* subsp. *veitchii*	*P. anomala* subsp. *veitchii*
MRC	1	2-19-0414	*P. suffruticosa*	*P. suffruticosa*	*P. suffruticosa*
	2	2-17-0559	*P. suffruticosa*	*P. suffruticosa*	*P. suffruticosa*
	3	2-16-0158	*P. suffruticosa*	*P. suffruticosa*	*P. suffruticosa*
	4	2-16-0138	*P. suffruticosa*	*P. suffruticosa*	*P. suffruticosa*
	5	2-14-0085	*P. suffruticosa*	*P. suffruticosa*	*P. suffruticosa*

^1^ Inconsistent results are indicated in bold.

## Data Availability

Not applicable.

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
