# Peer review of "Rapid Identification of Paeoniae Radix and Moutan Radicis Cortex Using a SCAR Marker-Based Conventional PCR Assay"

_plants, 2022, doi:10.3390/plants11212870_

Round 1
Reviewer 1 Report
1)I think the figure 3 is not clear. The information of boxes with different colors (red/blue/yellow/green) should be indicated.
2)It is better to show morphological characters of these four Paeonia species in a separate figure. It will highlight the importance of genetic identification for these medicinal plants visually.
3)Are the species-specific SCAR primers designed based on ITS or rbcL in Table 1?
4)Paeonia is an taxonomically intractable lineage. More species from this genus may be important to analyze the specificity of the developed SCAR primers in Table 2.
5)The language throughout the manuscript needs further smooth.
Author Response
Response to Reviewer 1 Comments
1)I think the figure 3 is not clear. The information of boxes with different colors (red/blue/yellow/green) should be indicated.
Response 1: Thank you for this comment. As suggested, we have corrected Figure 3. Line 145.
2)It is better to show morphological characters of these four Paeonia species in a separate figure. It will highlight the importance of genetic identification for these medicinal plants visually.
Response 2: Thank you for this comment. As suggested, we have added the morphological characteristics of the four Paeonia species to Figure S1. Line 33.
3)Are the species-specific SCAR primers designed based on ITS or rbcL in Table 1?
Response 3: Thank you for this comment. As commented, species-specific SCAR primers were designed based on ITS or rbcL sequences. We have provided more details in Table 1. Line 101.
4)Paeonia is an taxonomically intractable lineage. More species from this genus may be important to analyze the specificity of the developed SCAR primers in Table 2.
Response 4: Thank you for this comment. As commented, many Paeonia species are distributed worldwide. However, we performed this study on only four Paeonia species because these species are the only ones distributed as herbal medicines in the medicinal herb markets of South Korea. Because it is important to be able to distinguish between these species from a medicinal perspective and prevent adulteration, we developed SCAR markers and ITS DNA barcoding analysis for distinguishing between these Paeonia species. As the reviewer suggested, we intend to do a follow-up study to discriminate between many more Paeonia species.
5)The language throughout the manuscript needs further smooth.
Response 5: Thank you for this comment. As suggested, English language editing for the manuscript was performed twice by a professional proofreading company, Bioedit (http://www.bioedit.com).

Reviewer 2 Report
I believe the paper is interesting, and compatible to the journal.
Author Response
Response to Reviewer 2 Comments
I believe the paper is interesting, and compatible to the journal.
Response 1: Thank you for these comments.

Reviewer 3 Report
Dear authors,
I reckon this research has shown a good method to distinguish 4 different Paeonia species. The results look great, but there is still room for improvement. I do suggest adding more content and data to enrich the article. Please find my suggestions below:
- In line 11, please double-check the grammar for this sentence again.
- The abbreviation “SCAR” is used in the title and I expected it is important and the key to the paper. However, I cannot find it in the abstract and keywords. Please point out the importance of SCAR.
- In line 30, “Paeonia × suffruticosa“ is used as a scientific name. However, “Paeonia suffruticosa“ is written in your study materials. Just wondering which one is correct.
- Would you please add some more descriptions for PRR and PRA in line 34? And what is the difference between the two processing methods?
- In Tables 1 and 2, you have a target species called “P. anomala subsp. veitchii”. However, you use P. veitchii throughout the article. Just wondering which one is correct. Please make it consistence.
- In table 3, there are 3 different assays for sample identification. However, you only provided the result of SCAR. You must have more data to support morphological assay and DNA barcoding. For the morphological assay, you can provide pictures or drawings of different Paeonia species and point out the different species characters. For DNA barcoding, you should describe how to do it.
In my point of view, you need to demonstrate another evidence or assay data aligned with your SCAR assay and show the consistency with your SCAR assay. As you mentioned in the article, the reason that you want to identify the different species is due to the distinct bioactive compound composition. Therefore, I actually expected to see the bioactive compound assay aligned with your SCAR assay. If you are not able to analyze the bioactive compounds, you should present the data of the morphological assay and DNA barcoding assay at least. It is not convincing to me if you only present the SCAR assay data.
Author Response
Response to Reviewer 3 Comments
I reckon this research has shown a good method to distinguish 4 different Paeonia species. The results look great, but there is still room for improvement. I do suggest adding more content and data to enrich the article. Please find my suggestions below:
Response 1: Thank you for these comments.
In line 11, please double-check the grammar for this sentence again.
Response 2: Thank you for this comment. As suggested, we double-checked the sentence, but have decided to delete it and replace it with the following sentence: “Although there is a clear difference between the Korean and Chinese pharmacopeias of Paeoniae Radix, the processed roots of P. lactiflora and P. anomala subsp. veitchii are commonly used indiscriminately in the herbal market.” Lines 10-13.
The abbreviation “SCAR” is used in the title and I expected it is important and the key to the paper. However, I cannot find it in the abstract and keywords. Please point out the importance of SCAR.
Response 3: Thank you for this comment. As suggested, we have described the importance of SCAR. Lines 2-3, 20, 23.
In line 30, “Paeonia × suffruticosa“ is used as a scientific name. However, “Paeonia suffruticosa“ is written in your study materials. Just wondering which one is correct.
Response 4: Thank you for this comment. As suggested, we have corrected the scientific name of Paeonia suffruticosa. Lines 41, 131 (Table 2).
Would you please add some more descriptions for PRR and PRA in line 34? And what is the difference between the two processing methods?
Response 5: Thank you for this comment. As suggested, we have provided more information on the difference between PRR and PRA in the two processing methods in lines 36-40.
In Tables 1 and 2, you have a target species called “P. anomala subsp. veitchii”. However, you use P. veitchii throughout the article. Just wondering which one is correct. Please make it consistence.
Response 6: Thank you for this comment. As suggested, we have corrected the scientific name of P. anomala subsp. veitchii. Lines 7, 12, 17, 36, 53, 74, 89, 93, 98, 106, 114, 136, 148, 155, 156, 162 (Table 3), 175, 188, 238, 258.
In table 3, there are 3 different assays for sample identification. However, you only provided the result of SCAR. You must have more data to support morphological assay and DNA barcoding. For the morphological assay, you can provide pictures or drawings of different Paeonia species and point out the different species characters. For DNA barcoding, you should describe how to do it.
Response 7: Thank you for these comments. As suggested, we have added the sample pictures used in the morphological assay (Supplementary Figure 5) and the sequences of the ITS region used in the DNA barcoding assay (Supplementary Figure 6). Morphological identification was performed by a herbologist, a committee member of the Ministry of Food and Drug Safety (Material & Methods Section 4.5.). As you can see in Supplementary Figure 5, commercial herbal medicines are distributed in various forms, such as in slice and/or dried states. Therefore, in this study, we developed a genetic assay to compensate for the disadvantages of morphological identification. Furthermore, we have described in more detail that the DNA barcoding assay was carried out by performing sequence analysis of the ITS region in Table 3 and Material & Methods Section 4.2. Lines 155, 160.
In my point of view, you need to demonstrate another evidence or assay data aligned with your SCAR assay and show the consistency with your SCAR assay. As you mentioned in the article, the reason that you want to identify the different species is due to the distinct bioactive compound composition. Therefore, I actually expected to see the bioactive compound assay aligned with your SCAR assay. If you are not able to analyze the bioactive compounds, you should present the data of the morphological assay and DNA barcoding assay at least. It is not convincing to me if you only present the SCAR assay data.
Response 8: Thank you for these comments. As suggested, we have added the sample pictures used in the morphological assay (Figure 1 and Supplementary Figure 5) and the sequences of the ITS region used in the DNA barcoding assay (Supplementary Figure 6). Currently, although study of the bioactive compounds of Paeonia species is progressing, there have been no convincing reports about the precise bioactive compound composition of this species. Therefore, the purpose of our study is to offer an efficient and rapid species identification method based on a genetic assay to distinguish herbal medicines, which would be difficult using morphological identification alone.

Round 2
Reviewer 3 Report
Dear authors,
Thank you for your reply! It is great to have pictures that show the morphology in Figure S1 and also have more supplementary materials for table 2. There are several minor mistakes found in the article. I recommend accepting this paper after revision. Please find my suggestions below:
1. In line 96, “PJ-F (G → C)” was written, but the bold underlined character is T in table 1. I have checked the alignment data in Figure S2 and it seems not consistent. Please double-check the primer information in table 1, Figure S2, and Figure S3.
2. In Table 3, the subtitle “DNA Barcoding Assay of the ITS region” is not correct. For P. lactiflora, you used rbcL region as the description in line 259.
Please consider my suggestions before publishing. Although there is no bioactive compounds analysis In this article, I look forward to finding it in your future publication.
Author Response
Response to Reviewer 3 Comments
Dear authors,
Thank you for your reply! It is great to have pictures that show the morphology in Figure S1 and also have more supplementary materials for table 2. There are several minor mistakes found in the article. I recommend accepting this paper after revision. Please find my suggestions below:
Response 1: Thank you for these comments.
- In line 96, “PJ-F (G → C)” was written, but the bold underlined character is T in table 1. I have checked the alignment data in Figure S2 and it seems not consistent. Please double-check the primer information in table 1, Figure S2, and Figure S3.
Response 2: Thank you for this comment. As suggested, we have corrected “PJ-F (G→ T)” in line 96.
- In Table 3, the subtitle “DNA Barcoding Assay of the ITS region” is not correct. For P. lactiflora, you used rbcL region as the description in line 259.
Response 3: Thank you for this comment. Paeonia four species could be identified using the ITS DNA barcoding assay, but the ITS region was insufficient for the SCAR marker development in P. lactiflora. Therefore, we used the rbcL region in P. lactiflora for SCAR marker development. These paragraphs are described in Discussion lines 216-217 & 224-227.
Please consider my suggestions before publishing. Although there is no bioactive compounds analysis In this article, I look forward to finding it in your future publication.
Response 4: Thank you for these comments.
